# The Roles of *transformer-2* (*tra-2*) in the Sex Determination and Fertility of *Riptortus pedestris*, a Hemimetabolous Agricultural Pest

**DOI:** 10.3390/insects14110834

**Published:** 2023-10-24

**Authors:** Jinjun Ying, Haiqiang Wang, Biyun Wang, Zeping Mao, Youyuan Chen, Junmin Li, Chuanxi Zhang, Jichong Zhuo

**Affiliations:** State Key Laboratory for Managing Biotic and Chemical Threats to the Quality and Safety of Agro-Products, Key Laboratory of Biotechnology in Plant Protection of MARA and Zhejiang Province, Institute of Plant Virology, Ningbo University, Ningbo 315211, China; 2011074054@nbu.edu.cn (J.Y.); 2011074045@nbu.edu.cn (H.W.); biyun_wang@163.com (B.W.); 18268666546@163.com (Z.M.); chenyy1726@163.com (Y.C.); lijunmin@nbu.edu.cn (J.L.); chxzhang@zju.edu.cn (C.Z.)

**Keywords:** *Riptortus pedestris*, sex determination, *transformer-2*, fertility, hemimetabolous insects

## Abstract

**Simple Summary:**

Sex differentiation plays a vital role in the survival of any species with sexual reproduction. The sex determination pathway in holometabolous insects has been extensively researched and demonstrated to be both complex and diverse, and in particular, it has been shown that the primary signals are not conserved. However, the investigation of the sex-determining mechanisms in hemimetabolous insects remains limited and sparse, with only a few species having been studied thus far. One such species is the bean bug, *Riptortus pedestris*, a member of the Hemiptera order. This insect species is a primary cause of the typical soybean staygreen syndrome, a significant and pressing concern in soybean production. Therefore, in the present study, we focus on understanding the role of the *transformer-2* homolog in *R. pedestris* (*Rptra-2*) concerning sex determination and fertility in both sexes. Our findings contribute to a broader comprehension of the sex determination pathways in hemimetabolous insects and hold potential for future pest management strategies.

**Abstract:**

In most holometabolous insects, *transformer-2* (*tra-2*) is an auxiliary gene required for sex determination, exerting a crucial role in regulating sexual differentiation; however, the study of *tra-2* in hemimetabolous insects remains very sparse and limited to just a few species. In this study, we investigated the sequence and expression profile of the *tra-2* gene in the bean bug, *Riptortus pedestris*, an agricultural pest belonging to the Heteroptera order. Three non-sex-specific splicing isoforms of *Rptra-2* were found, *Rptra-2^293^*, *Rptra-2^284^*, and *Rptra-2^299^*, which shared most exons and exhibited similar expression throughout all stages of development, with particularly elevated levels in the embryo, ovary, and testis. RNAi knockdown experiments revealed that the suppression of *Rptra-2* in nymphs led to abnormal females, characterized the formation of male-specific external genital, and also caused longer nymph duration. Knockdown of the expression of the *Rptra-2* gene in newly emergent virgin females would cause ovarian arrest, and injecting the 8th-day virgin females with ds*Rptra-2* also caused a noticeable decline in the offspring numbers. Conversely, in ds*Rptra-2-*treated males, the testes maintained normal morphology but experienced impaired reproductive capacity, attributed to diminished sperm viability. These findings highlight the crucial role of *Rptra-2* in the sex determination and fertility of *R. pedestris*, providing valuable insights into the sex determination mechanisms of hemimetabolous insects.

## 1. Introduction

During their long evolutionary process, insects have developed complex and diverse sex determination pathways, which have been studied in many holometabolous insects, especially in *Drosophila melanogaster* [1,2,3,4,5,6]. The sex determination pathway of *D. melanogaster* is based on the sex-specific transcription of key regulatory genes, which organize in a cascade of X-linked signaling elements, (XSE) > *Sex-lethal* (*Sxl*) > *transformer* + *transformer-2* (*tra*/*tra-2*) > *doublesex* (*dsx*) [7,8]. In females, the functional SXL protein is activated by XSEs and regulates the splicing of *transformer* (*tra*) pre-mRNA to produce female-specific *tra* selective splicing, the functional TRA protein regulates the female-specific *dsx* mRNA selective splicing with the help of TRA-2, and then the female-specific *dsx* regulates the female-specific development of *D. melanogaster* [9,10]. In contrast, males are unable to form functional SXL proteins because they do not receive signals, producing nonfunctional male-specific TRA proteins, and thus *dsx* mRNA is spliced into the male form and regulates male-specific development. In females of *D. melanogaster*, the protein TRA-2 helps the TRA protein to bind on the exon splicing enhancer to regulate the sex-specific splicing of *dsx,* and is required for female sex determination in somatic cells; females develop into intersex individuals without *tra-2*. *Tra-2* is also required for spermatogenesis in male germ cells, and males suffer sperm defects, which lead to loss of fertility after *tra-2* knockdown.

Tra-2 function required for female sex determination is conserved in many insect species. The TRA-2 protein shows evolutionary conservation of an RNA recognition motif (RRM) and two arginine/serine (RS) domains [10,11]. In the medfly, *Ceratitis capitata*, another Diptera species, *tra-2* is not only required to regulate the sex-specific splicing of *Ccdsx*, but also involved in establishing and maintaining the autoregulation of *Cctra*, which is the key to female sex determination; moreover, a Y-link gene *Maleness-on-the-Y* (*MoY*) rather than an X-linked gene orchestrates sex determination in the early embryo stage [12]. Yeast two-hybrid assay shows that CcTRA interacts with CcTRA2. Interestingly, as has also been observed for *Drosophila* TRA2, CcTRA2 interacts with itself [13]. In the Oriental Fruit Fly, *Bactrocera dorsalis*, *tra-2* takes part in the sex determination pathway and has also been reported to play an important role in embryonic development. RNAi-mediated knockdown of *tra-2* generates male-only progeny, similarly to the RNAi-mediated knockdown of *tra* [14]. In the Hymenoptera species *Apis mellifera*, *tra-2* is an essential regulator for *dsx* female splice regulation. However, loss of *tra-2* functions causes embryo lethality in both sexes, which is unrelated to sex determination [15]. Moreover, in the lepidopteran *Bombyx mori*, the sex-specific splicing regulation of *dsx* does not require a *tra-2* orthologue [16]. In the Hemipteran *Nilaparvata lugens*, knockdown of *Nltra-2* in the embryo stage does not result in offspring sex bias, in which all female progenies develop into long-winged and males are short-winged [17], and they have also suggested that *Nltra-2* may play important roles in sex-specific splicing [18]. Moreover, in *N. lugens*, one new gene, *female determiner2* (*fmd2*), has been reported to cooperate with *female determiner* (*fem*) and be involved in the regulation of downstream gene *dsx*, which is essential for embryogenesis of both sexes [19]. Therefore, studying the functional diversity of *tra-2* provides a perspective to observe the complexity and diversity of sex-determining molecular mechanisms in insects.

*Riptortus pedestris* (Hemiptera: Coreidae) is an important agricultural pest with a wide geographical distribution, mainly in Asia, including China, South Korea, Japan, and India [20,21,22]. Recently, more attention has been paid to this pest, as field and laboratory experiments revealed that *R. pedestris* may be the main cause of soybean Zhengqing [22,23,24,25]. Zhengqing disease is a staygreen syndrome in soybeans that causes large soybean yield losses in northern China [22,26]. Zhang et al. used the Maxent model to predict the current and future distribution of *R. pedestris* under multidimensional environmental prediction factors (such as climate, land use, and elevation factors) [27]. They pointed out that in the future environmental scenario, the suitable habitat for *R. pedestris* has been significantly expanded, and there is a serious risk of *R. pedestris* outbreaks in soybean planting areas in China [27]. Currently, the main methods to control *R. pedestris* are aggregation pheromone traps and pesticides [28,29,30]; however these methods are inefficient [21,31]. Therefore, there is an urgent need for alternative strategies against *R. pedestris*. The sex determination genes have been reported to be used in insect pest management, in which the sex determination genes act as the target gene to produce all-male population collapse [14,32,33]. Regarding *Riptortus pedestris*, it has been reported that the presence of XX/X0 sex chromosomes may control its sex determination [34]. However, the genes responsible for sex determination in *R. pedestris* have not been reported. In this study, we have identified and characterized *Rptra-2*, the *tra-2* homologue of *R. pedestris*. Our findings aim to contribute a better understanding of the sex determination mechanisms in *R. pedestris* and potentially aid in addressing any future massive outbreaks.

## 2. Materials and Methods

### 2.1. Insect Rearing

The *R. pedestris* populations used here were originally collected in 2019 from soybean fields in Suzhou, China (33.7° N, 117.0° E). The insects were reared in cages (30 cm × 30 cm × 35 cm) by providing water and dried soybean seeds (Wandou 27), with water and seeds refreshed every three days. The bugs were reared at 26 ± 0.5 °C, with 16 L: 8 D cycles and around 50% relative humidity (RH).

### 2.2. Cloning of Rptra-2

The *tra-2* gene from the Hemiptera insect *Nilaparvata lugens* served as the search query to identify homologous genes in the *R. pedestris* transcriptome (https://ngdc.cncb.ac.cn/search/?dbId=gwh&q=GWHBAZH00000000, accessed on 15 July 2022) [17,35]. The full open reading frames (ORFs) from female and male complementary DNA (cDNA) libraries were amplified using the primers CDS-*Rptra2*-S and CDS-*Rptra2*- AS (Appendix A). The PCR products were cloned into the PMD-19T vector (TaKaRa, Dalian, China) and sequenced.

### 2.3. Sequence Analysis of Rptra-2

The exons and intron regions of *Rptra-2^293^*, *Rptra-2^284^*, and *Rptra-2^299^* were predicted using the Splign tool from NCBI (www.ncbi.nlm.nih.gov/sutils/splign/splign.cgi, accessed on 10 August 2022) by aligning the sequences of *Rptra-2^293^*, *Rptra-2^284^*, and *Rptra-2^299^* to the genomic sequence of *R. pedestris*. Amino acid sequence alignment of TRA-2 proteins was carried out for insect species from Diptera (*D. melanogaster*-AAF58232.2, *L. cuprina*-ACS34688.1, *A. aegypti*-AGW27097.1, *C. capitata*-NP_001266337.1, *A. amita*-CBJ17290.1), Hymenoptera (*A. mellifera*-NP_001252514.1), Lepidoptera (*B. mori*-XP_012553366.1), Coleoptera (*T. castaneum*-AHF71088.1), and Hemiptera (*N. lugens*-AWK28251.1, *B. tabaci*-QAB02877.1). The alignment was performed using Clustal X, version 2.1, with default parameters.

### 2.4. RNA Extraction

Different developmental stages and tissues of *R. pedestris* were homogenized in 1 mL of Trizol RNA extraction reagent (Thermo Fisher Scientific, Waltham, MA, USA). Samples were centrifuged at 12,000 rpm for 15 min at 4 °C. The solution was transferred to a 1.5 mL RNA-free centrifuge tube, chloroform was added, and it was incubated for 5 min at room temperature. The samples were then centrifuged at 12,000 rpm for 15 min at 4 °C. The supernatant was then transferred to a 1.5 mL RNA-free centrifuge tube. An equal volume of isopropanol was added to the supernatant, and the RNA was precipitated by centrifugation at 12,000 rpm for 15 min at 4 °C. The supernatant was discarded, and the ethanol was removed by washing twice with 1 mL of 75% ethanol. Finally, the precipitate was resuspended in RNase-free water after drying for 20 min on a clean bench.

### 2.5. Semiquantitative RT-PCR and Quantitative Real-Time PCR (qRT-PCR) Analysis

Total RNA was extracted from different development stages and tissues, and three independent biological replicates were performed. A total of 1 μg RNA was used to perform reverse transcription in a 10 μL reaction system using the HiScript II QRT SuperMix Kit (Vazyme, Nanjing, China) according to the manufacturer’s instructions, diluted 20 times, and 1 μL cDNA was used in subsequent PCRs. 

As previously described, the extracted total RNA was reverse transcribed. qRT-PCR was performed using the SYBR Green Supermix Kit (Yeasen, Shanghai, China) with the ABI QuantStudio 5 Real-Time PCR System (Thermo Fisher Scientific). The qRT-PCR primers, which were designed by Primer Premier 5.0, are listed in Appendix A. The *actin* gene of *R. pedestris* was used as the reference gene. The relative quantitative method (2^−∆∆Ct^) was used to evaluate quantitative variations [36].

### 2.6. Absolute Quantification in Real-Time qPCR

The cDNAs of whole insects obtained from previous reverse transcription were PCR-amplified using primers specific for *Rptra-2^284^* and *Rptra-2^299^*, respectively, and the resulting bands were purified. The concentration of the PCR products was measured with NanoDrop One/One_c_ ultra-micro spectrophotometer (Thermo Fisher Scientific) and diluted to 3.3 × 10^−2^ ng/μL and 3.5 × 10^−2^ ng/μL, respectively. Then the two PCR products were serially diluted seven times, each time by a factor of 10. The standard curves for *Rptra-2^284^* and *Rptra-2^299^* were constructed using the logarithm of the initial template copy number and the Ct value of the primers at each dilution. The values of the resulting straight lines are (Y = −3.249 ∗ X + 27.28 and Y = −3.154 ∗ X + 25.33) and the values of the R-squares (R^2^ = 0.9951 and R^2^ = 0.9867), respectively. The copy number of the PCR products was calculated separately using the following equation: $$\mathrm{DNA}\;\left(\mathrm{copy}\right)=\frac{6.02\times 10^{23}\left(\mathrm{copy}/\mathrm{mol}\right)\times \mathrm{DNA}\;\mathrm{amount}\left(g\right)}{\mathrm{DNA}\;\mathrm{length}\;\left(\mathrm{bp}\right)\times 660\;\left(g/\mathrm{mol}/\mathrm{dp}\right)}$$

Refs. [37,38] obtained Ct values of dilutions by real-time PCR using *Rptra-2^284^* and *Rptra-2^293^* specific primers. The PCR amplification efficiency (E) was calculated for both primer pairs and the E values were found to be in the range of 90% to 110%.

### 2.7. dsRNA Synthesis and Injection

Primers (Appendix A) with the T7 promoter were designed to synthesize double-stranded RNA (dsRNA) targeting the ORF sequences of *Rptra-2* and *gfp*, respectively. The dsRNA lengths were 359 and 378 bp, respectively. dsRNA was prepared using the MEGAscript T7 High Yield Transcription Kit (Thermo Fisher Scientific) according to the manufacturer’s instructions. The NanoDrop One/One_c_ ultra-micro spectrophotometer (Thermo Fisher Scientific) was used to quantify the concentration of the product. A total of 500 ng/μL of dsRNAs was injected into the thorax of third-instar nymphs using the Eppendorf electric microinjector FemtoJet 4i (Eppendorf, Hamburg, Germany). The settings of the FemtoJet 4i were air pressure: 1000, interval: 10, sub pressure: 10. Needle pulling used capillaries at 58 °C. A total of nine nymphs were used as samples for RNA isolation three days after injection. The ds*gfp*-treated nymphs were used as controls in all injection experiments. To observe the growth and development of nymphs, 50 nymphs in the third instar were injected with dsRNAs, and RNAi-treated individuals were reared until adult emergence. The phenotypes of the control insects and treatment insects were compared under a Nikon SMZ25 full-automatic stereoscopic fluorescence microscope (Nikon, Tokyo, Japan). Three biologically independent replicates were performed.

### 2.8. Dissection and Fertility Analysis

The newly emerged adults were collected every 12 h, and the virgin females and males were injected with ds*gfp* and ds*Rptra-2*, respectively. After being reared for 9 days, ovaries and testes were dissected, and phenotypes were observed and photographed under a Nikon SMZ25 full-automatic stereoscopic fluorescence microscope (Nikon, Tokyo, Japan). In the fertility analysis experiments, the newly emerged adults were injected with dsRNAs, respectively. A female and a male adult were separately reared on soybean seeds in a plastic cup for mating, and the number of eggs was examined 20 days after mating. At least 10 pairs of adults per treatment were tested. 

### 2.9. Sperm Viability Test

To test sperm viability, the newly emerged males were injected with dsRNAs, and the males were reared alone for 9 days before dissecting the spermatophores. Sperm activity was measured according to the manufacturer’s protocol using the Live/Dead Sperm Viability Kit (L-7011) (Invitrogen, Carlsbad, CA, USA). The vas deferens was cut with tweezers in 1 mL HEPES-buffered saline solution containing bovine serum albumin (10 mM HEPES, 150 mM NaCl, 10% BSA, pH 7.4) to release semen. Then, 5 μL of diluted SYBR14 dye was added to a 1 mL sample of diluted semen and incubated at 36 °C for 15 min. Next, 5 μL of propidium iodide (Component B) was added to the 1 mL sample of diluted semen and incubated for another 10 min. Finally, the number of green sperm (live) and red sperm (dead) in 10 μL of samples was counted using the Leica TCS SP8 X confocal microscope (Leica, Wetzlar, Germany). Each semen sample represented five individual insects, and each treatment was pooled from three independent experiments.

### 2.10. Maternal RNAi

A total of 100–150 fifth-instar nymphs were placed in nylon cages (30 cm × 30 cm × 35 cm) and raised with soybean seedlings. The newly emerged adults were collected every 12 h, and the females and males were reared separately. After being reared for 8 days, the virgin females were injected with ds*gfp* and ds*Rptra-2*, respectively, and then mated with wild-type males. Every pair of male and female was allowed to oviposit on soybean seedlings for 20 days. The number of eggs laid was counted every day, and the eggs were moved to the cages for feeding. The number of hatched nymphs was recorded until offspring adults emerged. Offspring adults from each mating treatment were separated for the sex ratio calculation. A total of 20 to 30 biological replicates were performed for each mating pair treatment.

### 2.11. Mating Experiment

Adult males and females were reared separately within 12 h of emergence, and dsRNAs were injected into the newly emerged males. Nine days later, the virgin females were placed in plastic dishes (90 mm diameter, 15 mm high) at the same time as each of the unmated control or treated males. Observation was continued for two hours and the success of courtship, frequency of courtship, mating start time, and duration of mating were recorded. Mating is considered successful if the duration of copulation is more than 60 min. The observations were made during the photophase in a laboratory kept at 26 °C and 50% RH. 

### 2.12. Statistical Analysis

All data analyses were performed using GraphPad Prism, version 8.0.2 (GraphPad Software Inc., San Diego, CA, USA). Differences in the mean and standard error of the mean (SEM) among the groups in the RNAi study were analyzed by analysis of variance. For one-to-one comparisons, Student’s *t*-test was used to compare the statistical significance between each treatment. The level of significance has been determined by the following *p*-values: * *p* < 0.05, ** *p* < 0.01, *** *p* < 0.001, while ns indicates no significant difference.

## 3. Results

### 3.1. Structure and Conservation of Rptra-2

The *R. pedestris tra-2* sequence (GenBank: BAN20641.1) was reported in 2012; they uploaded the intestinal transcriptome of *R. pedestris* [39]. In order to determine whether other *tra-2* transcripts exist in *R. pedestris* and their specific functions, we employed the protein sequence of *Nilaparvata lugens* TRA-2 (NlTRA-2) (AWK28252.1) as a query to conduct a blast analysis on the transcriptome of *R. pedestris* [17,35]. Consequently, *Rptra-2* was successfully identified in *R. pedestris*, and its cDNA sequence was subsequently compared to the genome sequence. The *Rptra-2* gene yielded three distinct splice variants, namely *Rptra-2^28^*^4^ (OR058869), *Rptra-2^293^* (OR058870), and *Rptra-2^299^* (OR637360), characterized by seven introns and eight exons (Figure 1A). These transcripts exhibited lengths of 852, 879, and 897 bp, and encoding proteins comprising 284, 293, and 299 amino acids, respectively. Upon alignment of the *Rptra-2^284^*, *Rptra-2^293^*, and *Rptra-2^299^* sequences, minor discrepancies (27 nucleotides) were observed in exon 4 and encoded nine amino acids (KMKKYFGTQ), as well as a 1515 bp skip in exon 8 in *Rptra-2^299^* (Figure 1A). TRA2 and its homologs in many other insects belonged to the Ser-Arg-rich (SR) protein family, which has an RNA recognition motif (RRM) with two ribonucleoprotein regions (RNP1 and RNP2) followed by the linker region and flanked by two arginine/serine-rich domains (RS1 and RS2) (Figure 1B). In this study, we found that RpTRA-2^293^, RpTRA-2^284^, and RpTRA-2^299^ shared the same protein sequences in these conserved domains, suggesting that RpTRA-2^293^, RpTRA-2^284^, and RpTRA-2^299^ might play the same functions in the sex determination of *R. pedestris* (Figure 1B). 

### 3.2. Expression Profile of Rptra-2

As three Tra-2 isoforms were identified in *R. pedestris*, we investigated the expression levels of *Rptra-2^284^* and *Rptra-2^299^* by absolute quantification (specific primers for *Rptra-2^293^* could not be designed due to the lack of specific fragments distinguishing it from the other two transcripts). The results showed that *Rptra-2^284^* and *Rptra-2^299^* were present in both sexes. However, the expression of *Rptra-2^299^* was four to five times higher than that of *Rptra-2^284^* in both females and males (Appendix A).

To investigate the spatial expression pattern of *Rptra-2* in various tissues, including heads, muscles, salivary glands, gut, epidermis, ovaries, and testes, we employed semi-quantitative PCR and qRT-PCR techniques. The primers used targeted the common sequences of *Rptra-2^293^*, *Rptra-2^299^*, and *Rptra-2^284^*. As a control for expression, the constitutive gene *Rpactin* was utilized. The results showed that *Rptra-2* was expressed in all tissues and that the relative transcript levels of *Rptra-2* were significantly higher in the ovaries and testes than in other tissues (Figure 2A,B). Additionally, we examined the expression patterns of *Rptra-2* at different developmental stages of *R. pedestris*, including eggs, first- to fifth-instar nymphs, and female and male adults. The results revealed that *Rptra-2* exhibited expression throughout all developmental stages and in both female and male adults, suggesting its potential significance in development. The relative transcript level of *Rptra-2* peaked during the embryo stage and gradually declined after 36 h (Figure 2C,D). Semi-quantitative PCR with products that covered the 27-nucleotide gap between *Rptra-2^293^* and *Rptra-2^284^* was employed in this study. The results demonstrated the expression of both transcripts in all developmental stages and tissues (Appendix A). For a more accurate comparison, q-PCR was used to detect the spatiotemporal expression profiles of *Rptra-2^284^* and *Rptra-2^299^* by specific primers for *Rptra-2^284^* and *Rptra-2^299^*, respectively. The results also showed that both transcripts were expressed at all developmental stages and in different tissues with similar expression (Appendix A).

Importantly, the outcomes revealed the presence of *Rptra-2* in all development stages and tissues, with notably higher expression in the early embryo stages, ovaries, and testes, suggesting that *Rptra-2* might play critical roles in the development of *R. pedestris* (Figure 2).

### 3.3. Nymphs’ RNA Interference Revealed the Function of Rptra-2 in Sex Determination

To elucidate the role of *Rptra-2* in the sex determination of *R. pedestris*, we injected *Rptra-2*-specific dsRNA, targeting the common sequences of *Rptra-2^284^*, *Rptra-2^293^*, and *Rptra-2^299^*, into third-instar larvae. This injection resulted in a significant reduction in *Rptra-2* mRNA levels 72 h post-injection (Figure 3A). Following adult emergence, we observed that the male-to-female ratio did not show a significant difference compared to the control group (Figure 3C); however, there was a delay in the development time of females, suggesting that *Rptra-2* played a critical role in the female development of *R. pedestris* (Figure 3B). Males subjected to ds*Rptra-2* treatment did not exhibit apparent alterations in their external genitalia. More than 70.50% of RNAi-treated females exhibit varying degrees of genital malformations. Distinctions between male and female adults can typically be made based on their external genitalia. In many female specimens treated with ds*Rptra-2*, genital malformations and the occurrence of male-specific clasper were documented (Figure 4B), indicating that the gene *Rptra-2* influenced the female-specific somatic development of *R. pedestris*, rather than males.

### 3.4. Rptra-2 Influenced R. pedestris Fertility in Both Sexes 

As *Rptra-2* is highly expressed in both the ovary and testis, we aimed to investigate its critical role in the fertility of *R. pedestris*. In this study, we injected newly emerged adults (0–12 h) with dsRNAs and dissected them after 9 days of rearing, to examine the impact of *Rptra-2* on the internal genital structures of *R. pedestris*. The findings revealed that the ovaries in ds*Rptra-2* females were underdeveloped compared to ds*gfp* females (Figure 5A,B), while no noticeable differences were observed in the testes of ds*Rptra-2* males compared to ds*gfp* males (Figure 5C,D). To assess the fecundity of *R. pedestris* and determine the number of eggs laid by females within 20 days, we observed that all females were unable to produce offspring (Table 1). Subsequently, upon mating ds*Rptra-2* males with ds*gfp* virgin females, we observed a reduced number of eggs laid compared to the control group (Table 1). However, the eggs failed to hatch and exhibited desiccation (Figure 5E,F). And we further studied the mating behavior of ds*Rptra-2*-treated males and found that there was no obvious difference with control ones, except that the mating success rate of ds*Rptra-2*-treated males was lower (44%) (Appendix A). Therefore, we performed the sperm viability test and found that the majority of sperm from ds*Rptra-2* males had diminished viability, which led to the males losing fertility (Figure 5G,H). These results highlight the essential role of *Rptra-2* in the fertility of *R. pedestris*.

### 3.5. Maternal RNAi Knockdown of Rptra-2

Transcripts of *Rptra-2* were detected during the early embryo stage, indicating its maternal contribution. To investigate the functions of *Rptra-2* in embryo development, we injected 8-day-old virgin females with ds*Rptra-2* and ds*gfp*, respectively. The efficiency of maternal RNAi of *Rptra-2* was evaluated by examining the eggs, nymphs, and adults produced in the mating experiment. The results demonstrated a significant decrease of 96.61% in *Rptra-2* expression in eggs laid by females treated with ds*Rptra-2*, with a gradual and slow recovery observed (Figure 6A). The number of offspring laid by RNAi females was counted from each pair on the 20th day after mating initiation. It was found that the number of nymphs laid by *Rptra-2* RNAi females was significantly lower compared to control females (Figure 6B). Nymphs were raised until emergence, and subsequent calculation of sex segregation showed that the offspring sex ratio of *Rptra-2* RNAi females was similar to that of control females (Figure 6C).

The study results indicate that *Rptra-2* plays a critical role in female fertility, while embryonic *Rptra-2* knockdown does not affect offspring sex ratio.

## 4. Discussion

In the sex determination pathway of insects, *tra-2* has been reported to play a critical role in most species. In this study, we identified the multiple functions of *tra-2* homologs in the soybean pest *R. pedestris* (*Rptra-2*), one hemimetabolous insect. *Rptra-2* produces three non-sex-specific splice isoforms, *Rptra-2^284^*, *Rptra-2^293^*, and *Rptra-2^299^*, whose sequences differ only somewhat in exon 4 and exon 8 (Figure 1A). RpTRA-2^293^, RpTRA-2^284^, and RpTRA-2^299^ share almost the same protein sequences, in which the conserved domains locate, and their expression in all developmental stages and tissues are also similar. Therefore, *Rptra-2^284^*, *Rptra-2^293^*, and *Rptra-2^299^* might share the same functions in *R. pedestris*, and we use two dsRNAs targeting the same sequences to study the functions of *Rptra-2* in this study (Appendix A). 

Sexual development, one of the most significant and widely studied developmental processes, is regulated by sex-determining cascade. In the sexual differentiation of *D. melanogaster*, the TRA-2 and TRA complex plays a crucial role in regulating the female-specific alternative splicing of *dsx*. When *tra-2* is absent, the *dsx* pre-mRNA is spliced in a male-specific isoform, resulting in the development of intersex individuals, whose anal plates and sex combs are intermediate between those of males and females [40]. In *Ceratitis capitata*, with RNAi-mediated knockdown of *tra-2* in the embryo stage, the females exhibit an intersexual phenotype with typical female traits but male genitalia [12]. In the Hemiptera species *N. lugens*, RNAi-mediated knockdown of the expression of *tra-2* in the earlier nymph stage resulted in a higher degree of virilization of females [17]. In *Bemisia tabaci*, silencing *Bttra2* causes male genital (anal) malformations. It does not affect the female phenotype, but reduces the expression of the female vitellogenin gene [41]. In this study, RNAi-mediated knockdown of *Rptra-2* in third-instar nymphs led to the development of females with genital abnormalities and the presence of male-specific claspers, while males developed normally (Figure 4B). Our findings suggest that *Rptra-2* is involved in the somatic female-specific development of *R. pedestris*. Furthermore, the relationship between *Rptra-2* and *dsx* homologs warrants further investigation in future studies.

In this study, we observed a high expression of *Rptra-2* in the reproductive systems, specifically in the context of ovary development. Through RNAi-mediated knockdown of *Rptra-2* in newly emerged females, we demonstrated that ovarian development was inhibited, resulting in female infertility. In *D. melanogaster*, tra-2 is needed by females for yolk protein synthesis, which is important for ovary development [42]. In *Aedes albopictus*, RNAi-mediated knockdown of two *tra-2* splicing isoforms suppressed ovarian development significantly, and the transcription of the yolk protein gene was reduced [43]. However, it remains unclear whether *Rptra-2* influences ovarian development in *R. pedestris* by regulating yolk protein synthesis. Future research is necessary to elucidate this aspect. Additionally, we discovered that treating male *R. pedestris* with ds*Rptra-2* resulted in infertility due to the death of a majority of sperm, despite no significant morphological changes observed in the testes (Figure 5). The role of *transformer-2* in spermatogenesis was first reported in *Drosophila melanogaster* [44], and in *Nilaparvata lugens*, 80% of the spermatozoa of males injected with ds*Nltra-2* were also found to be inactive [17]. These findings underscore the essential role of *Rptra-2* in the fertility of both sexes. We also found that the mating success rate of ds*Rptra-2*-treated males decreased obviously compared with the control ones, and we also found that knocking out *tra-2* in female nymphs leads to an extension of developmental time, suggesting that the influence of *Rptra-2* on the development of *R. pedestris* is different across sexes (Figure 3B). 

In this study, we used maternal RNAi to silence *Rptra-2* in the early embryo stage, and the RNAi interference could last to the third-instar offspring (Figure 6A). In this case, knockdown of *Rptra-2* in early embryos resulted in decreased numbers of offspring, and no sex bias was found in the offspring (Figure 6B,C). A significant increase in the proportion of male offspring compared to controls after silencing *tra-2* has been previously reported in many insects as well as in Hymenoptera *A. mellifera* and Coleoptera *T. castaneum* [15,45]. In *C. capitata*, a *Tra-2^TS2^* mutant population was constructed using CRISPR/Cas9 technology to successfully achieve 100% female to male conversion [32]. However, no change in offspring sex ratio was observed in the Hemiptera *Nilaparvata lugens* and *Asian citrus psyllid* after embryonic RNAi knockdown of *tra-2* [17,46], indicating that *tra-2* homologs in Hemiptera species might have different functions in embryogenesis. Moreover, in *N. lugens*, the new feminizing switch *fmd* rather than *tra* has been reported to regulate the sex-specific alternative splicing of *dsx*, in which a new cooperator gene *fmd2* was reported to be involved, suggesting that the molecular mechanism of *tra-2* homologs involved in the Hemiptera species sex determination was different from holometabolous insects [19]. In order to further study the functions of *Rptra-2* in *R. pedestris* sex determination, more work needs to be carried out in the future, such as studying whether *fmd/fmd2* or *tra* homologs regulate the downstream gene *dsx*, or what molecular mechanisms of *Rptra2* are involved in the sex determination of *R. pedestri*. Gene editing is already feasible in numerous Hemipteran pests. Presently, we are exploring appropriate methods for editing the genes of *R. pedestris*, which may provide more insights into the functions of *Rptra-2* in future [47].

## Figures and Tables

**Figure 1 insects-14-00834-f001:**
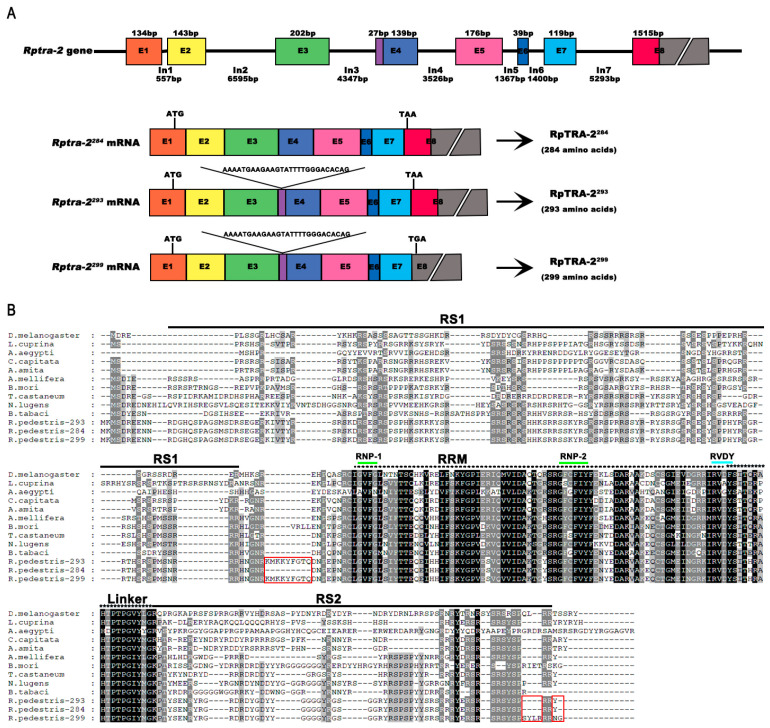
Gene structure of *Rptra-2* and alignment of TRA-2 proteins. (**A**) The gene structure of the *Riptortus pedestris transformer-2* (*Rptra-2*) gene shows three alternatively spliced variants and their deduced proteins. The square boxes labeled E1–E8 represent exons, and the black lines (In1–In7) between these boxes are introns. The number under each intron represents the number of nucleotides. The potential start codons (ATG) and stop codons are indicated. (**B**) Amino acid sequence alignment of TRA-2 proteins of insect species from Diptera, Hymenoptera, Lepidoptera, Coleoptera, and Hemiptera. Different background colours are given depending on the level of conservatism, with black being the most conservative, grey being the second most conservative, and very low conservatism being shown in white. The RS1 and RS2 domains, containing arginine and serine, are represented by single lines, while the RNA recognition motif (RRM) is represented by a broken line. Furthermore, two regions of ribonucleoproteins (RNP-1 and RNP-2) are represented by green lines, the blue line indicates a conserved RVDY motif and the linker region is depicted by asterisks (*). Red boxes are used to highlight varying amino acids between different transcripts of RpTRA-2.

**Figure 2 insects-14-00834-f002:**
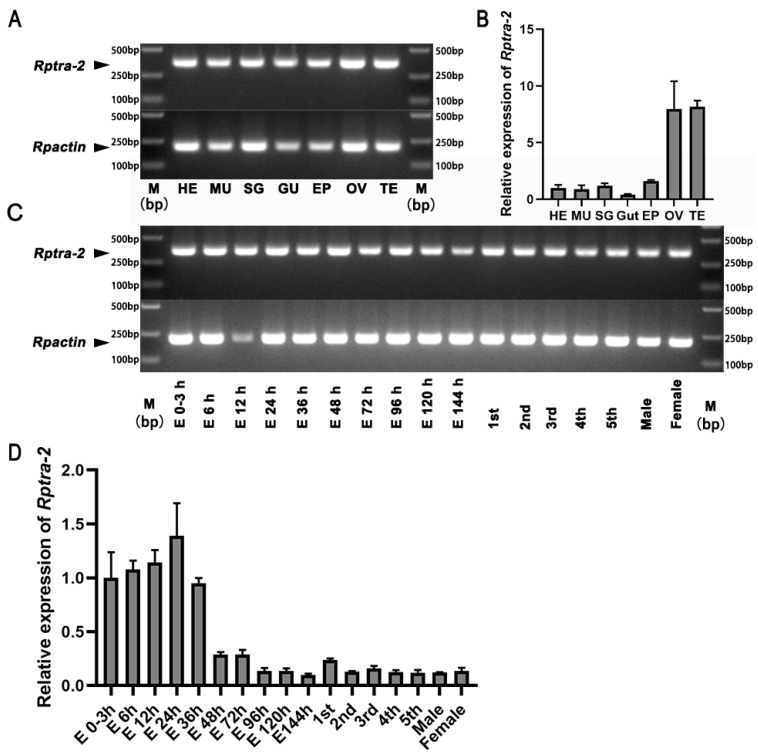
Temporal and spatial expression profiles of *Rptra-2*. (**A**,**B**) Detection of *Rptra-2* in different tissues of *R. pedestris* adults (HE, head; MU, muscle; SG, salivary glands; GU, gut; EP, epidermis; OV, ovaries; TE, testes) by RT-PCR and qRT-PCR (*n* = 5). (**C**,**D**) Detection of *Rptra-2* in different developmental stages, including eggs (*n* = 20); different larval instars: first (*n* = 8), second (*n* = 6), third (*n* = 5), fourth (*n* = 3), fifth instar (*n* = 3); male adults (*n* = 3); and female adults (*n* = 3) by RT-PCR and qRT-PCR. The *Rpactin* gene was used as a reference gene to normalize *Rptra-2* gene expression level. All treatments were independently replicated three times.

**Figure 3 insects-14-00834-f003:**
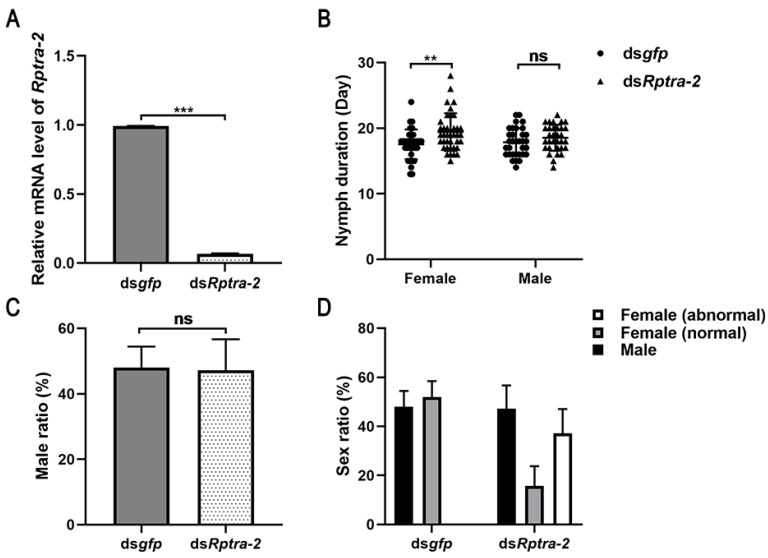
Influence of *Rptra-2* on nymph duration and sex ratio. (**A**) Knockdown efficiency. qRT-qPCR was performed on RNA collected from 3 whole *R. pedestris* at 3 days post-injection. Ct values for *Rptra-2* were normalized to the *Rpactin* gene, and *Rptra-2* expression was compared between *gfp-* and *Rptra-2*-silenced *R. pedestris* using delta–delta Ct (ΔΔCt) method for relative gene expression. Student’s *t*-test was used, *** *p* < 0.001. (**B**) Time to become females and males in freshly molted third-instar nymphs treated with ds*gfp*/ds*Rptra-2*, respectively. Student’s *t*-test was used, ** *p* < 0.01, ns = non-significant. (**C**) The male ratio of third-instar ds*gfp*-treated nymphs (*n* = 3) is shown in grey columns, and the male ratio of third-instar ds*Rptra-2*-treated nymphs (*n* = 3) is shown in white columns. Student’s *t*-test was used, ns = non-significant. (**D**) Sex ratio of third-instar ds*gfp*-treated nymphs and ds*Rptra-2*-treated nymphs after emergence. The number of males is shown in black columns, the number of normal females is shown in grey columns, and the number of females with genital malformations is shown in white columns.

**Figure 4 insects-14-00834-f004:**
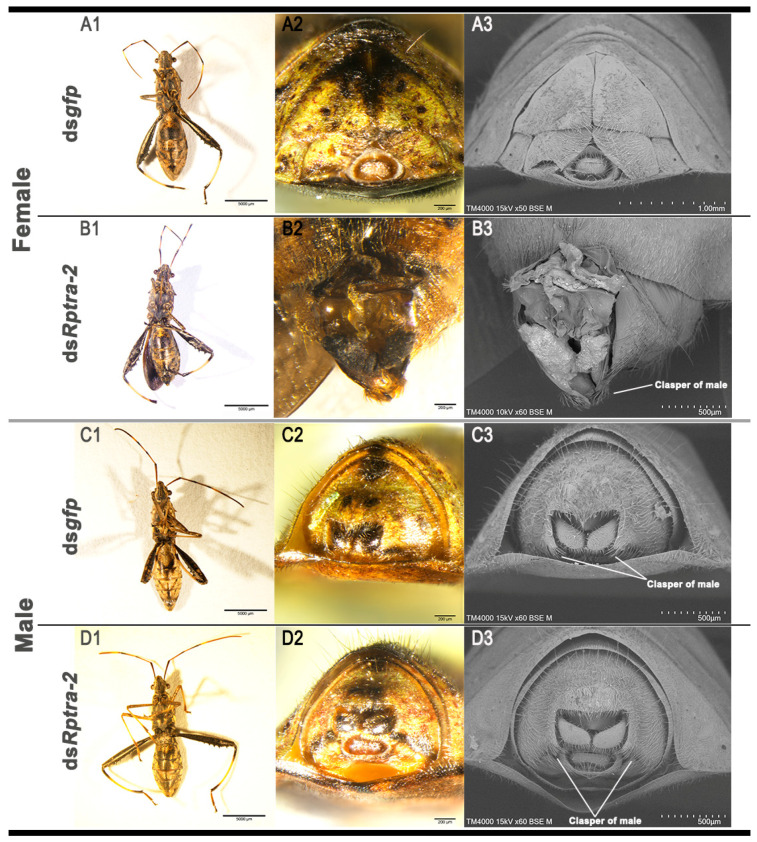
Phenotypes of females and males after ds*gfp* and ds*Rptra-2* injections at the third-instar stage. (**A1**–**A3**,**B1**–**B3**) The phenotypes and external genitalia of adult females injected with ds*gfp* (**A1**–**A3**) or ds*Rptra-2* (**B1**–**B3**) at the third-instar stage. (**C1**–**C3**,**D1**–**D3**) The phenotypes and external genitalia of adult males injected with ds*gfp* (**C1**–**C3**) or ds*Rptra-2* (**D1**–**D3**) at the third-instar stage. All insects were collected on the six days after emergence. The clasper of male is labeled.

**Figure 5 insects-14-00834-f005:**
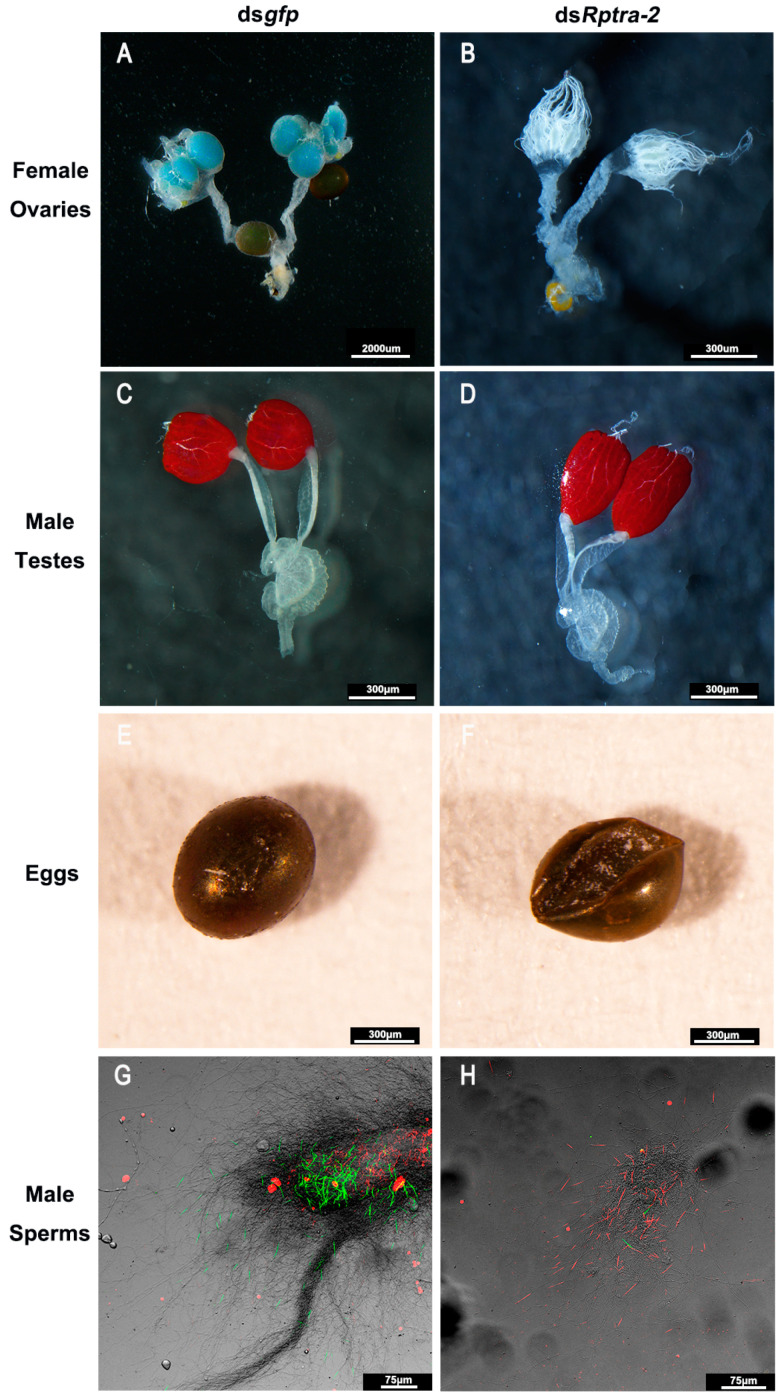
The effect of ds*Rptra-2* on the development of ovaries and testes. (**A**,**B**) Ovaries of ds*gfp-* and ds*Rptra-2*-treated virgin females 9 days after emergence. (**C**,**D**) Testes of ds*gfp-* and ds*Rptra-2*-treated males 9 days after emergence. (**E**,**F**) Eggs from ds*gfp-* and ds*Rptra-2*-treated males mating with wild-type females, respectively. (**G**,**H**) Sperm from ds*gfp* and ds*Rptra-2* male adults 9 days after emergence; green indicates live sperm and red indicates dead sperm.

**Figure 6 insects-14-00834-f006:**
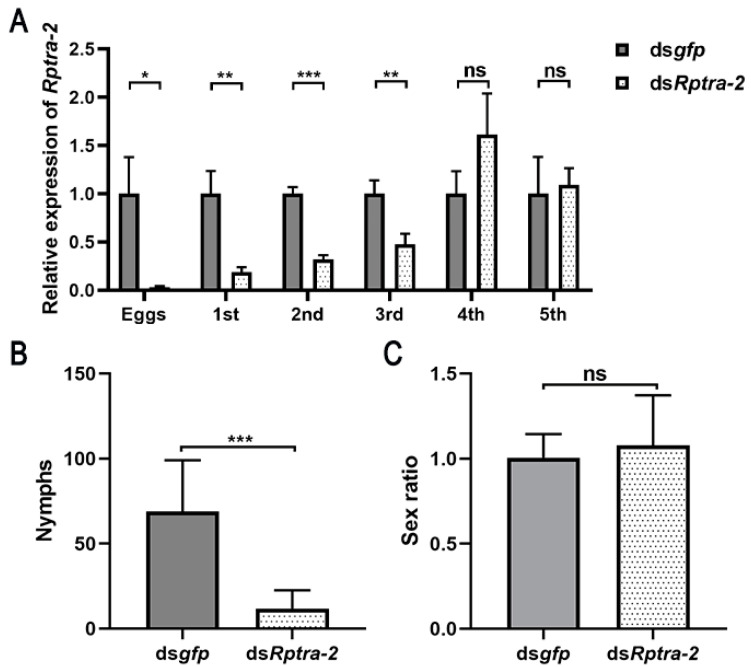
The influence of maternal RNAi silencing of *Rptra-2*. (**A**) The efficiency of maternal ds*Rptra-2* on the development of the offspring: eggs (*n* = 15), first (*n* = 8), second (*n* = 6), third (*n* = 5), fourth (*n* = 3), and fifth instar (*n* = 3). The relative expression of *Rptra-2* in different instars of the offspring of ds*gfp* females (gray) and ds*Rptra-2* females (white). Student’s *t*-test was used (* *p* < 0.05, ** *p* < 0.01, *** *p* < 0.001, ns indicates no significant difference). (**B**) The number of offspring from ds*gfp* (*n* = 29) and ds*Rptra-2* females (*n* = 23). Student’s *t*-test was used, *** *p* < 0.001. (**C**) The male/female ratio of offspring. The male/female ratio of offspring from ds*gfp*-treated females (*n* = 10) is shown in grey columns, and the male/female ratio of offspring from ds*Rptra-2*-treated females (*n* = 10) is shown in white columns. Student’s *t*-test was used, ns indicates no significant difference.

**Table 1 insects-14-00834-t001:** Influence of *Rptra-2* on female and male fertility.

Treatment	Eggs	Dead Eggs	Nymph
ds*gfp* female × ds*gfp* male (*n* = 15)	113 ± 21	9 ± 4	104 ± 19
ds*Rptra-2* female × ds*gfp* male (*n* = 13)	0	0	0
ds*gfp* female × dsR*ptra-2* male (*n* = 12)	53 ± 13	49 ± 13	4 ± 3

## Data Availability

Sequences of *Rptra-2^284^* and *Rptra-2^293^* were deposited in GenBank with accession numbers OR058869, OR058870, and OR637360, respectively.

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
