# Peer review of "The Roles of transformer-2 (tra-2) in the Sex Determination and Fertility of Riptortus pedestris, a Hemimetabolous Agricultural Pest"

_insects, 2023, doi:10.3390/insects14110834_

Round 1

Reviewer 1 Report

Comments and Suggestions for Authors

This research paper evaluates the role of transformer-2 (tra-2) in sex determination and fertility in a hemimetabolous insect, Riptortus pedestris. After the identification of Rptra2 by sequence alignment, its expression is measured by RT-PCR during development and in organs. The functions of Rptra2 were then tested using a loss of function strategy with RNAi injection at different time of development to evaluate the impact on sex determination, sex organs development and fertility, and finally early embryonic contribution. The presented results convincingly show that Rptra2 is required for somatic female-specific development, for ovaries development and female fertility. In addition, the authors present data on the role of Rptra2 in spermatogenesis.

The experiments appear to be well conducted and sufficiently controlled. The experimental designs are appropriate and the conclusions are well supported. However, even though the authors study the functions of Tra2 in an hemimetabolous insects, Riptortus pedestris in which it was not studied before, I must confess that according to me, even though the experiments are well conducted and convincing, the novelty on tra2 function is somehow limited even for this species. For instance, the partner of tra2, tra is never tested as a control for example. Splicing forms of downstream targets such as dsx are not evaluated (it is only mentioned in the discussion as future work). This would have been interested to validate the loss of function by RNAi injection. And the role of tra2 in spermatogenesis was already described in Drosophila melanogaster nearly 40 years (Belotte and Baker, Dev Biology, 1983). This reference was omitted in the introduction and should be added.

Overall, I accept the paper with minor revisions. But I would really appreciate if the authors can perform quantifications of the sex-specific isoforms of dsx in their loss of function approaches. This would strengthen the quality of their data and would be a additional validation of their experimental approach.

Minor comments:

Some English corrections and spelling mistakes should be corrected to improve the quality of the text.

- line 66-68: “In Hymenoptera species, Apis mellifera, tra-2 is an essential regulator for dsx female splice regulation, however, lost the functions of tra-2 67 causes embryo lethality in both sexes, which is unrelated to sex determination.”

- “actain” instead of “actin” can be found in the main text, the material and methods section and the figure legends.

- Line 410 : “….that the influences of Rptra-2 on the development of R. pedestris are systematically.” I am unsure of what it means.

- Figure 4 could be improved: the images should be shown with the same orientation (column 2 and 3). And figure legends are insufficient and has to completed.

Comments on the Quality of English Language

Some English corrections and spelling mistakes should be corrected to improve the quality of the text.

Some English corrections and spelling mistakes should be corrected to improve the quality of the text.

- line 66-68: “In Hymenoptera species, Apis melliferatra-2 is an essential regulator for dsx female splice regulation, however, lost the functions of tra-2 67 causes embryo lethality in both sexes, which is unrelated to sex determination.”

- “actain” instead of “actin” can be found in the main text, the material and methods section and the figure legends.

- Line 410 : “….that the influences of Rptra-2 on the development of R. pedestris are systematically.” I am unsure of what it means.

And I am guessing I missed many other spelling mistakes.

Reviewer 2 Report

Comments and Suggestions for Authors

Ying et al., submitted a paper titled “The role of transformer-2 (tra-2) in the sex determination and fertility of Riptortus pedestris, a hemimetabolous agricultural pest”.

They presented novel data in this hemipteran pest insect, showing that the tra-2 gene has conservation in its structure and possibly also in its function. They investigated the expression profile during development and in dissected tissues, showing high expression in the reproductive organs. 

The paper can contribute to the knowledge of the field after a major revision of the text, the addition of few informatics analyses (NlFmd conservation and dsx conservation), an RTpCR additional analysis on putative Rpdsx sex-specific isoforms (strongly suggested), following RNAi, and also a qPCR on a second Rptra-2-related gene which RNAi could target (see below), to exclude that RNAi effects are not due to the specific targeting another gene having similarity to Rptra2.

Comments on the Quality of English Language

The MS is written clearly, but it can be slightly improved.

Round 2

Reviewer 2 Report

Comments and Suggestions for Authors

I am glad that the authors appreciated the suggestions and improved the clarity and solidity of MS significantly.

I would be very glad if the authors would upload their DNA sequence dataset also at NCBI and not only on a Chinese database. They also took advantage of the  NCBI sequence databases for their work, as every biology scientist in our world, and we should continue to maintain the same collaborative space, I think.

line 145: the PCR on DNA or cDNA? What does "recovered" mean? Specify the steps performed (e.g., RNA extraction, reverse transcription, RT-PCR, purification of the band obtained, quantification of the purified product, etc.)

Line 146-148: this part is unclear. How many 10-fold dilutions did you perform? What is the starting quantity in nanograms (or copy number)? Is this base 10 logarithm (LOG10)?

Line 154-155: you should report the exact value of E, the values of the line obtained (y=ax + b), intercept and slope, and finally also the value of R squared.

Line 161-162: "DNA" or "cDNA", because you previously talked about RNA extractions and reverse transcription.

Line 164: unnecessarily repetitive

Line 168: instead of "internal control" you could say "reference gene".

Still to be corrected

ovaries and testes were dissected and phenotypes were observed and photographed un-192 der a Nikon SMZ225 full-automatic stereoscopic fluorescence microscope (Nikon, Japan). 193 

ovaries and testes were dissected and phenotypes were observed and photographed un-192 der a Nikon SMZ25 

Figure 5. 

Please add larger length bars as reference within each photo.

Comments on the Quality of English Language

I suggest to ask for professional English editing as some sentences need still corrections. 

Here as examples, a list of sentences which require professional English editing (also in there style).

 Simple Summary: 

12-  especially the primary signals are extremely unconservative 

collapse [14,32,33]. With regard to Riptortus pedestris, the presence of XX / X0 sex chromo-96 somes has been reported and suggested as a possible control of its sex determination. 97 Insects 2023, 14, x FOR PEER REVIEW 3 of 16  However, the sex determination pathway of R. pedestris has not been reported [34]. In this 98 

critical for the fertility of both sexes, and we hope our results could help us to learn more 101 knowledge about the sex determination mechanism of R. pedestris and provide more tar-102 gets to suppress possible future mass outbreaks of it. 103 

2.2. Cloning of Rptra-2 113 

The tra-2 gene from Hemiptera insect Nilaparvata lugens was used as a query to search 114 for homologous genes in the R. pedestris transcriptome of our yet unreported and reported 115 (https://ngdc.cncb.ac.cn/search/?dbId=gwh&q=GWHBAZH00000000) [17,35]. The full 116 

RNP-2) regions are indicated by green lines, and the linker region is indicated by asterisks (*). Differential amino  304 

acids between different transcripts of RpTRA-2 are indicated by red boxes. 305 

In lots of dsRptra-2 treated females, 317 genital malformations and the presence of male-specific clasper were observed (Figure 318 

3.4. R. pedestris became infertile following Rptra-2 mRNAreduction by RNAi 

3.4. R. pedestris became infertile following Rptra-2 mRNA reduction by RNAi 

These results showed that Rptra-2 is required for female’s fertility, but knockdown of 375 embryonic Rptra-2 doesn’t alter offspring sex ratio. 37

Gene editing is already possible in many Hemipteran pests 467 [47], but we have not yet established an effective CRISPR/Cas9 system in our lab for Riptor-468 tus pedestris, we will continue to explore effective gene editing techniques for Riptortus 469 pedestris. , 
